# The Arf-GAP Proteins AoGcs1 and AoGts1 Regulate Mycelial Development, Endocytosis, and Pathogenicity in *Arthrobotrys oligospora*

**DOI:** 10.3390/jof8050463

**Published:** 2022-04-29

**Authors:** Le Yang, Xuemei Li, Yuxin Ma, Keqin Zhang, Jinkui Yang

**Affiliations:** State Key Laboratory for Conservation and Utilization of Bio-Resources in Yunnan, Key Laboratory for Southwest Microbial Diversity of the Ministry of Education, School of Life Science, Yunnan University, Kunming 650032, China; yangle@mail.ynu.edu.cn (L.Y.); xmli@mail.ynu.edu.cn (X.L.); mayuxin0521@163.com (Y.M.); kqzhang1@ynu.edu.cn (K.Z.)

**Keywords:** *Arthrobotrys oligospora*, Arf-GAP protein, conidiation, mitochondrial activity, stress response, pathogenicity

## Abstract

Small GTPases from the ADP-ribosylation factor (Arf) family and their activating proteins (Arf-GAPs) regulate mycelial development, endocytosis, and virulence in fungi. Here, we identified two orthologous Arf-GAP proteins, AoGcs1 and AoGts1, in a typical nematode-trapping fungus *Arthrobotrys oligospora*. The transcription of *Aogcs1* and *Aogts1* was highly expressed in the sporulation stage. The deletion of *Aogcs1* and *Aogts1* caused defects in DNA damage, endocytosis, scavenging of reactive oxygen species, lipid droplet storage, mitochondrial activity, autophagy, serine protease activity, and the response to endoplasmic reticulum stress. The combined effects resulted in slow growth, decreased sporulation capacity, increased susceptibility to chemical stressors and heat shock, and decreased pathogenicity of the mutants compared with the wild-type (WT) strain. Although deletion of *Aogcs1* and *Aogts1* produced similar phenotfypic traits, their roles varied in conidiation and proteolytic activity. The Δ*Aogts1* mutant showed a remarkable reduction in conidial yield compared with the WT strain but not in proteolytic activity; in contrast, the Δ*Aogcs1* mutant showed an increase in proteolytic activity but not in sporulation. In addition, the growth of Δ*Aogcs1* and Δ*Aogts1* mutants was promoted by rapamycin, and the Δ*Aogts1* mutant was sensitive to H-89. Collectively, the Δ*Aogts1* mutant showed a more remarkable difference compared with the WT strain than the Δ*Aog**cs1* mutant. Our study further illustrates the importance of Arf-GAPs in the growth, development, and pathogenicity of nematode-trapping fungi.

## 1. Introduction

The superfamily of small G-proteins (GTPases) that mediate signal transduction in eukaryotic cells can be classified into at least five subfamilies, including Rho, Rab, Ras, Ran, and Arf [1,2]. These proteins are known for their prime roles in vesicle trafficking and actin cytoskeleton remodeling [3]. These superfamilies serve as hydrolases that bind and hydrolyze GTP to form GDP and act as molecular switches for small G-proteins [4]. The cycling of small GTPases is accomplished by switching between an activated state (GTP-bound state) and an inactive state (GDP-bound state), and GTPase-activating proteins (GAPs) are important proteins involved in this transition. Upon the action of GAPs, the hydrolase activity of the small GTPase itself is activated, and the phosphate group is released, thereby hydrolyzing the activated small GTPase into an inactive state [5].

ADP ribosylation factors (Arfs) belong to the small GTPases subfamily and interact with many proteins and other molecules that regulate their state of activation. The function of an Arf depends on switching between inactive Arf-GDP and active Arf-GTP. Thus, the function of an Arf depends on the guanine nucleotide exchange factor and the Arf-GAP that drives the GTP-binding and hydrolysis cycle [6]. Arfs were originally discovered due to their ability to stimulate the ADP-ribosyltransferase activity of the cholera toxin [7,8]. Subsequent studies have shown that Arfs are master regulators of membrane trafficking responsible for many important processes, including exocytosis, endocytosis [9], and cell differentiation [10]. Arfs also play a role in the formation of the Golgi-endoplasmic reticulum (ER) system and in the loss of proteins during transportation in the cytoplasm to supplement membrane surface proteins in the ER [11]. Therefore, Arf-GAPs are a family of proteins that induce GTP hydrolysis bound to Arfs and are proposed to regulate the disassembly and dissociation of vesicle coats [12].

The Arf GTPase subfamily in yeast includes five Arf-GAPs, namely, Glo3, Gcs1, Gts1, Age1, and Age2. Gcs1 is thought to have a more critical role in yeast than other Arf-GAPs due to the fact that knockdown of the gene encoding Gcs1 is highly lethal [13]. The role of Gcs1 in yeast was determined to activate the intrinsic activity of Arf1, Arf1, and Arl1 as well as participating in intracellular exocytosis and vesicular trafficking pathways [9]. Colocalization studies have shown that Gcs1 is mainly present in the Golgi and intracellular compartments, and it is required to maintain mitochondrial morphology and stimulate actin polymerization in the actin cytoskeleton [14,15,16]. In *Candida albicans*, the *age3* gene, which is homologous to the yeast *gcs1* gene, plays a role in the biological process of efficient endocytosis, and a lack of *age3* results in defective hyphal growth, loss of invasive hyphae, and attenuated drug resistance [17]. In the plant pathogenic fungus *Magnaporthe oryzae*, MoGlo3 is required for vegetative growth, sporulation, and pathogenicity. Additionally, it is involved in reactive oxygen species (ROS) degradation, ER pressure response, vesicle formation, and maintenance of the Golgi complex [18]. Knockout of *gts1* severely impairs hyphal growth, delays uptake of the lipophilic dye FM4-64, and causes defective endocytosis in the cotton-infecting pathogen *Ashbya gossypii* [19]. At present, the roles of Arf-GAPs in filamentous fungi are still largely unclear.

Nematophagous fungi are a special group that can attack nematodes by capturing, producing toxins, and parasitizing. Among them, nematode-trapping (NT) fungi were the first type of nematophagous fungi to be studied. They hunt and kill nematodes through predatory organs (traps) such as adhesive three-dimensional nets, adhesive balls, adhesive branches, and contractile rings specially formed by vegetative hyphae [20,21]. As the main natural enemies of nematodes in soil ecosystems, NT fungi play a crucial role in regulating the balance of nematode populations in the ecosystem and, therefore, they have special ecological significance [22]. *Arthrobotrys oligospora* is the model strain of NT fungi, since a three-dimensional net can be generated to capture nematodes when there is a target host in the environment [20,23,24,25]. Extracellular serine proteases are significant pathogenic factors in *A. oligospora* and other NT fungi during their infection of nematodes [26,27,28]. Since the genome of *A. oligospora* was sequenced in 2011, studies have focused on regulating mycelial growth, development, and pathogenicity of *A. oligospora* [29,30,31,32,33,34,35].

Recently, an Arf-GAP Glo3 was characterized in *A. oligospora*, and it plays a crucial role in growth, conidiation, endocytosis, trap formation, and pathogenicity [19]. To further explore the functions of other Arf-GAPs in this fungus, two orthologous Arf-GAPs, AoGcs1 and AoGts1, were retrieved, and their functions in the growth and development of *A. oligospora* were identified by gene knockout and multi-phenotypic analysis.

## 2. Materials and Methods

### 2.1. Fungal Strains and Culture Conditions

*Arthrobotrys oligospora* (ATCC24927) purchased from the American Type Culture Collection (ATCC) and deletion mutants generated during this study were routinely cultured on potato dextrose agar (PDA) medium at 28 °C. The *Saccharomyces cerevisiae* FY834 strain was grown in yeast extract-peptone-dextrose (YPD) broth, and SC-Ura medium was used to screen recombinant transformants [36]. *Escherichia coli* DH5α was used to clone the constructed vector and grown in LB medium at 37 °C. Oat medium was used to grow *Caenorhabditis elegans* (strain N2) at 26 °C in the dark for a week. PDA, tryptone glucose (TG), and tryptone yeast extract glucose agar (TYGA) were used for the phenotypic analysis as previously described [37] and corn-maizena yeast extract (CMY) was used for sporulation as previously described [38].

### 2.2. Sequence Analysis

The homologous proteins of Gcs1 and Gts1 were retrieved from the fungus *A. oligospora*, and the amino acid sequences of AoGcs1 (AOL_s00078g496) and AoGts1 (AOL_s00076g212) were downloaded from GenBank. The pI/MW tool (http://web.expasy.org/compute_pi/, accessed on 16 December 2021) [39] was used to calculate the theoretical isoelectric point (pI) and molecular weight (MW). Subsequently, its conserved functional domains were analyzed using InterProScan (http://www.ebi.ac.uk/Tools/pfa/iprscan/, accessed on 16 December 2021) [40]. A neighbor-joining phylogenetic tree for orthologous Gcs1 and Gts1 from diverse fungi were constructed using the MEGA 7 software package [41].

### 2.3. Deletion of Aogcs1 and Aogts1

The replacement fragments for the target gene were constructed as previously described [42]. Briefly, the 5′ and 3′ DNA sequences of the target genes were amplified from *A. oligospora* using paired primers (Appendix A), and the hygromycin resistance gene (*hph*) cassette was amplified from pCSN44. Next, all fragments including the 5′ sequence, 3′ sequence, *hph* sequence, and the linearized pRS426 plasmid digested with *Eco*RI and *Xho*I were transformed into the *S. cerevisiae* FY834 strain [43]. Then, the yeast plasmid with AoGcs1-hph-pRS426 and AoGts1-hph-pRS426 were isolated and used as DNA templates, and the gene replacement fragment was amplified using paired primers (Appendix A). Finally, the replacement fragments were chemically introduced into *A. oligospora* protoplasts and spread on PDAS medium with 200 μg/mL hygromycin B [44]. Each transformant for genes *Aogcs1* and *Aogts1* was verified by a PCR method using the specific primers YZ-F and YZ-R (Appendix A) as previously described [42], and the positive transformants were further identified by Southern blot analysis, which was performed using the North2South Chemiluminescent Hybridization Detection Kit (Pierce, Rockford, IL, USA).

### 2.4. Comparison of Vegetative Growth and Conidia

The wild-type (WT) and mutant strains were cultured on PDA medium for 5 days, then 7 mm discs were collected from each WT and mutant strain using a hole puncher and inoculated onto PDA, TG, and TYGA medium for incubation at 28 °C for 7 days to determine the colonial diameter for each fungal strain. To assess the sporulation of each strain, 7 mm equal-sized fungal colonies from each strain were inoculated in the center of the CMY medium nad incubated at 28 °C for 14 days as previously described [45]. The number and morphology of conidia and conidiophores were observed by the side-shot method on the WA plate [46,47]. Conidial germination rates were determined as previously described [45]. Briefly, equal amounts of conidia from the WT and mutant strains were incubated on MM medium at 28 °C, and germinated conidia were calculated at specific time points (i.e., 4, 8, and 12 h). The experiment was repeated three times for each strain.

Additionally, conidia of the WT and mutant strains were stained for 30 min with 10 μg/mL calcofluor white (CFW) and 10 μg/mL 4′,6-diamidino-2-phenylindole (DAPI) then the morphology of the spores was observed using a fluorescence microscope (Leica, Mannheim, Germany).

### 2.5. Analysis of DNA Damage and Cell Apoptosis

Colonies from the WT and mutant strains were cultured on PDA with sterile coverslips for 7 days at 28 °C, and then the nuclei and morphology of mycelia were stained with DAPI and CFW, respectively. DNA damage and apoptosis were assessed by the terminal deoxynucleotidyl transferase-mediated dUTP nick end-labeling (TUNEL) assay [48]. The nuclei of the hyphae were stained with propidium iodide (PI), and fragmented DNA in the hyphae was re-stained with FITC-dUTP. The ratio of green to red fluorescence intensities was determined for at least 30 fields under a microscope to quantify the extent of DNA damage.

### 2.6. Comparison of Stress Tolerance

Seven millimeter fungal discs from WT and mutant strains were cultured on TG medium (control) supplemented with different concentration gradients of chemical stressors, including osmotic stressors NaCl (0.1, 0.2, and 0.3 M) and sorbitol (0.25, 0.50, and 1 M), oxidative stressor H_2_O_2_ (5, 10, and 15 mM), menadione (0.01, 0.03, and 0.05 mM), or the cell wall stress agents sodium dodecyl sulfate (SDS) (0.01, 0.02, and 0.03%) and Congo red (0.02, 0.05, and 0.1 mg/mL). Inhibitors of the mTOR and cAMP signaling pathway, rapamycin (10 µM) and H-89 (10 µM), were used at 28 °C for 7 days to test the response of the WT and mutant strains to intracellular signals [29]. In addition, fungal mycelium discs of the same size (i.e., 7 mm) from different strains were inoculated onto TG medium and incubated at different temperatures (28, 34, 38, and 42 °C) for 8 h and then returned to 28 °C until the seventh day to examine the sensitivity to heat shock [49]. Colonial morphology was inspected, the diameter of each colony was measured, and the relative growth inhibition (RGI) value was calculated as previously described [29,50]. The experiment was repeated three times for each strain.

### 2.7. Trap Formation, Nematocidal, and Proteolytic Activity

Approximately 1 × 10^5^ conidia from each strain were spread on WA plates and incubated at 28 °C for 3 days. Then, 200 nematodes were added to the center of each plate to induce trap formation and to observe and quantify the number of traps and captured nematodes at 12 h intervals [51]. The experiment was repeated three times for each strain.

To determine proteolytic activity, equal-sized fungal colonies were added to PD broth supplemented with 20 mL skim milk (5%) and cultured at 180 rpm at 28 °C for 7 days. The fermentation liquids were collected in an Erlenmeyer flask and the protease activity was evaluated on casein plates as previously described [52]. Quantitative analysis of protease activity was performed as previously described [52,53].

### 2.8. Analyses of Lipid Droplets (LDs), Endocytosis, ROS Accumulation, and Autophagy

To observe LDs, each strain was incubated on PDA plates with sterile coverslips at 28 °C for 7 days. Then, the mycelia were treated with 30 µL of 10 µg/mL boron dipyrromethene (BODIPY) dye at 37 °C in the dark, and LDs were observed by fluorescence microscope. The morphology of LDs was also observed by transmission electron microscope (TEM; Hitachi, Tokyo, Japan). To compare the endocytosis of fungal strains, the mycelial samples were stained with FM4-64, and the changes in FM4-64 at 0 s, 10 s, 1 min, and 3 min after entering the hyphae were observed by fluorescence microscope.

In addition, the fluorescent dyes dihydroethidium (DHE) and monodansylcadaverine (MDC) were used at a concentration of 10 µg/mL to analyze the accumulation of ROS and autophagy in hyphae, respectively. Subsequently, autophagy activity was analyzed by estimating the fluorescence intensity values of MDC observed under a fluorescence microscope for at least 30 fields, and ROS levels were estimated from the intensity of DHE fluorescence. Meanwhile, autophagosomes were observed by TEM.

### 2.9. Analysis of ER Stress and Mitochondrial Activity

Hyphal discs (7 mm) were cultured on TG medium supplemented with 10 mM dithiothreitol (DTT) and 10 mM tunicamycin for 7 days at 28 °C to test the ER stress response of WT and mutant strains. Then, the colony diameter was measured, and the RGI value was calculated [29,50]. For the analysis of mitochondrial morphology and mitochondrial membrane potential (MMP), hyphal discs were cultured on PDA plates with a glass slide. Afterwards, the hyphae were treated with 10 µg/mL MitoTracker Green or 10 µg/mL tetraethylbenzimidazolyl-carbocyanine iodide (JC-1) dye and incubated in the dark for 30 min, and images were collected with a fluorescence microscope. The ratio of green/red fluorescence intensity was calculated using ImageJ software (https://imagej.nih.gov/ij, accessed on 16 January 2022) for at least 30 randomly selected fields of view per slide. The ratio of green to red fluorescence intensities of JC-1 staining was used to assess MMP levels [29].

### 2.10. Quantitative Real-Time PCR (RT-PCR)

To analyze transcription levels for the genes involved in sporulation, cell wall synthesis, oxidative stress, cAMP signaling, and autophagy at different stages of growth and development in *A. oligospora*, the WT and mutant strains were cultivated on TYGA medium at 28 °C for 3, 5, and 7 days, and hyphae were collected for RT-PCR analysis. In addition, the WT strain was induced with nematodes for 0, 12, 24, 36, and 48 h, and hyphae were collected to determine transcription levels of the Aogcs1 and Aogts1 genes during trap formation. The total RNA from all mycelia samples was isolated using an RNA Extraction Kit (Axygen, Suzhou, China) and reverse-transcribed cDNA was prepared using a FastQuant RT Kit (Takara, Kusatsu, Japan). The transcription levels for candidate genes were then analyzed by RT-PCR assays with specific paired primers (Appendix A). The β-tubulin gene (AOL_s00076g640) from *A. oligospora* was used as an internal standard. RT-PCR was performed in triplicate for each gene as previously described [54]. The transcription levels for the target genes were calculated using the 2^−ΔΔCt^ method; briefly, fold changes were calculated using the following formula: 2^−ΔΔCt^, where ΔΔCt is ΔCt (treatment) − ΔCt (control), ΔCt is Ct (target gene) − Ct (tub), and Ct is the threshold cycle [55]. In addition, the relative transcript level (RTL) for each gene was quantified as the transcript level of each mutant versus WT.

### 2.11. Statistical Analyses

GraphPad Prism version 9.3.1 (GraphPad Software, San Diego, CA, USA) was used for photographic and statistical analysis. Data from triplicate experiments were examined using a one-way analysis of variance (ANOVA) and ranked by Tukey’s multiple comparison test. *p* < 0.05 was considered a significant difference.

## 3. Results

### 3.1. Sequence Analysis, Phylogenetic Relationship, and Transcription of Aogcs1 and Aogts1

Two Arf-GAP proteins, namely, AoGcs1 and AoGts1, were retrieved from *A. oligospora* using the homologous sequences of *S. cerevisiae* as queries. AoGcs1 and AoGts1 were found to contain 366 and 622 amino acid residues with predicted isoelectric points of 7.30 and 10.16 and molecular weights of 39.2 and 72.8 kDa, respectively. Gcs1 and Gts1 contained a conserved Arf-GAP domain in different fungi, and Gts1 also contained a ubiquitin-associated (UBA) domain (Appendix A). AoGcs1 shared a high degree of similarity to orthologs from the NT fungi *Dactylellina haptotyla* (81.7%) and *Duddingtonia flagrans* (93.7%), and it shared a middle degree of similarity (44.8–47.9%) to orthologs from other filamentous fungi; similarly, AoGts1 shared a high degree of similarity to orthologs from NT fungi *D. haptotyla* (75.5%) and *D. flagrans* (95.5%), and it shared a similarity of 28.3–31.7% to orthologs from other filamentous fungi (Appendix A). The phylogenetic tree of homologous Gcs1 and Gts1 from different fungi was constructed, and they were separated into two clades (Appendix A). The transcription of *Aogcs1* and *Aogts1* were downregulated during trap formation (compared to zero time), but significantly upregulated (*p* < 0.05) during the conidiation stage (from 3 to 9 days) compared with those in the vegetative growth stage (day 2) (Appendix A).

### 3.2. AoGcs1 and AoGts1 Regulate Vegetative Growth and DNA Damage

The genes *Aogcs1* and *Aogts1* were deleted (Appendix A) and two mutants of gene *Aogcs1* and three mutants of gene *Aogts1* were obtained for further study. Two mutants (i.e., Δ*Aogcs1* and Δ*Aogts1*) showed significantly reduced mycelial growth (*p* < 0.05) on PDA, TG, and TYGA media (Figure 1A), and the average colony diameters for the WT, Δ*Aogcs1*, and Δ*Aogts1* strains on PDA medium were 8.3, 6.5, and 4.8 cm, respectively. The corresponding colony diameters for each strain when grown on TG medium were 8.5, 5.2, and 4.9 cm compared to 8.5, 6.5, and 5.1 cm when grown on TYGA medium (Figure 1B). The colonial morphology of the Δ*Aogts1* mutant was irregular (Figure 1A). After the mycelia and cell nuclei of the WT and mutant strains were stained with CFW and DAPI, the mycelia of the Δ*Aogts1* mutant were more swollen and irregular than the WT strain (Figure 1C). Additionally, the nuclei of the mutants were difficult to stain as they appeared diffused and fragmented (Figure 1C). Fragmentation of chromosomal DNA was detected in the mycelia of the mutant strains using the TUNEL assay, and the nuclei were in a diffuse form when stained with PI (Figure 1C). Since most of the DNA in the mutants was in a diffuse state, the FI ratios of FITC-dUTP and PI were calculated to analyze the differences in the extent of DNA damage and cell apoptosis between the WT and mutant strains. The FI ratios for the WT, Δ*Aogcs1*, and Δ*Aogts1* strains were 0.09, 0.33, and 0.55, respectively (Figure 1D).

### 3.3. AoGcs1 and AoGts1 Regulate Conidiation, Conidial Morphology, and Conidial Germination

The WT strain produced many conidiophores, and multiple spores formed in the denticle node of the conidiophore. However, few conidiophores were observed for the Δ*Aogts1* mutant (Figure 2A). The spore yields for the WT, Δ*Aogcs1*, and Δ*Aogts1* were 30.9, 28.8, and 9.7 × 10^5^ spores/mL, respectively (Figure 2C). After staining the spore morphology and cell nucleus with CFW and DAPI, the conidia produced by the WT strain were usually obovoid in shape and formed one septum near the base, whereas most of the conidia from the Δ*Aogcs1* (48%) and Δ*Aogts1* (63%) mutants did not feature a septum or a change in morphology (Figure 2B,D). However, no difference in conidial germination of the Δ*Aogcs1* mutant was observed when compared with the WT strain. At 4 h, approximately 73.8%, 77.3%, and 57.8% of spores had germinated in the WT strain, Δ*Aogcs1*, and Δ*Aogts1*, respectively (Figure 2E). Compared with the WT strain, the transcript levels of all five genes involved in sporulation, namely, *abaA*, *veA*, *velB*, *vosA*, and *fluG*, were downregulated on day 5 in the Δ*Aogcs1* mutant. In addition, expression levels of *abaA*, *velB*, *vosA*, and *fluG* were also downregulated in the Δ*Aogts1* mutant on day 5. In contrast, the expression of *abaA*, *vosA*, and *fluG* was significantly upregulated (*p* < 0.05) in the Δ*Aogcs1* mutant on day 7. Furthermore, the expression of *veA* was significantly upregulated (*p* < 0.05) in the Δ*Aogts1* mutant at all tested time points, whereas the expression levels of other genes were upregulated at only some time points (Figure 2F).

### 3.4. AoGcs1 and AoGts1 Are Involved in Resistance to Chemical Stresses

The WT strain and mutants were compared for their responses to different chemical agents (Figure 3A and Appendix A). Compared with the WT strain, the RGI values for the Δ*Aogcs1* mutant were significantly increased (*p* < 0.05) in the presence of 0.1 M NaCl, 0.25 M sorbitol, 0.03% SDS, 20 ng/mL Congo red, and any concentration of H_2_O_2_ (5–15 mM) and menadione (0.01–0.05 mM). Similarly, the RGI values for the Δ*Aogts1* mutant significantly increased (*p* < 0.05) in the presence of 0.1–0.3 M NaCl, 0.01% SDS, 0.01–0.03 mM menadione, and any concentration of sorbitol (0.25–1 M), Congo red (20–100 ng/mL), and H_2_O_2_ (5–15 mM) (Figure 3B–E and Appendix A).

To probe the regulatory mechanism of Arf-GAP proteins in stress resistance, the transcription levels of fifteen putative genes involved in oxidative stress, hyperosmolarity, and cell wall synthesis were analyzed. The expression of five genes involved in osmotic pressure was downregulated in the Δ*Aogcs1* mutant on day 5. However, the expression of all five genes was upregulated on day 3 and day 7, especially the level of the *msn2* gene (*p* < 0.05). Similar to the observations for the Δ*Aogcs1* mutant, the expression levels of three genes, namely, *sho1*, *sln1*, and *ssk1*, were downregulated in the Δ*Aogts1* mutant on days 3 and 5, and all genes were significantly upregulated on day 7 (*p* < 0.05) (Figure 3F). Transcription levels of five genes involved in cell wall synthesis (i.e., *chs1*, *chs2*, *gfpa*, *gls*, and *glu*) were analyzed in mutants and WT strains, and all five genes were downregulated in the Δ*Aogcs1* mutant on day 5. Similarly, the transcript levels of four genes (i.e., *chs1*, *gfpa*, *gls*, and *glu*) were also downregulated on day 5 in the Δ*Aogts1* mutant, whereas the *chs2* gene was upregulated. In contrast, the expression of *chs1*, *chs2*, *gfpa*, and *gls* was significantly upregulated (*p* < 0.05) in the Δ*Aogts1* mutant on days 3 and 7. Similarly, the expression of *chs2* and *gfpa* was also significantly upregulated (*p* < 0.05) in the Δ*Aogcs1* mutant on days 3 and 7 (Appendix A). When the expression of five genes related to oxidative stress, namely, *per*, *thr*, *trxR*, *trx*, and *glr*, was analyzed, the results showed that the *thr* gene was significantly downregulated (*p* < 0.05) in the Δ*Aogcs1* mutant at all tested time points, whereas the expression levels of the other genes were downregulated at only some time points. The expression of two genes, *trx* and *glr*, was significantly downregulated (*p* < 0.05) in the Δ*Aogts1* mutant at all tested time points, and the expression levels of the other three genes were downregulated on day 5. In contrast, the expression of three genes (i.e., *trxR*, *trx*, and *glr*) was upregulated in the Δ*Aogcs1* mutant on days 3 and 7. Noticeably, the expression of *trxR* was significantly upregulated (*p* < 0.05) in the Δ*Aogts1* mutant on days 3 and 5, implying that this gene could be important for the ability of AoGts1 to regulate the fungal response to oxidants (Figure 3G).

### 3.5. AoGcs1 and AoGts1 Participate in Trap Formation, Nematocidal Activity, and Proteolytic Activity

After being induced with nematodes, the WT strain produced approximately 121 traps per cm^2^ at 48 h, whereas the Δ*Aogcs1* and Δ*Aogts1* mutants produced approximately 104 and 82 traps per cm^2^, respectively (Figure 4A,B). Correspondingly, the nematocidal activity of Δ*Aogcs1* and Δ*Aogts1* was also reduced. The WT strain captured 14.7%, 56.4%, 93.3%, and 96.8% of nematodes at 12, 24, 36, and 48 h, respectively. Relative to the WT, the Δ*Aogcs1* mutant captured 5.6%, 46.9%, 78.9%, and 92.7% of nematodes at the same four time points, respectively, compared to 2.9%, 26.0%, 51.8%, and 84.7% of nematodes for the Δ*Aogts1* mutant (Figure 4C).

Moreover, compared with the WT strain (49.8 U/g), disruption of the *Aogts1* gene (45.5 U/g) had little influence on the proteolytic activity of the mutant. In contrast, the Δ*Aogcs1* mutant (71.7 U/g) displayed increased proteolytic activities (Figure 4D,E). The transcriptional levels of five serine protease genes in *A. oligospora* were determined using RT-PCR. Of these, *54g992*, *75g8*, *78g136*, and *215g702* were significantly downregulated (*p* < 0.05) in the Δ*Aogts1* mutant on days 3 and 5, whereas the transcriptional levels of those genes were significantly upregulated (*p* < 0.05) on day 7. For the Δ*Aogcs1* mutant, the expression levels of all five genes were significantly downregulated (*p* < 0.05) on day 5 compared to WT, whereas very few genes were downregulated at the other time points (Figure 4F).

### 3.6. AoGcs1 and AoGts1 Play an Important Role in Heat Shock, and the ΔAogts1 Mutant Is Sensitive to H89

The heat shock response of the WT, Δ*Aogcs1*, and Δ*Aogts1* strains was determined, and the growth rates of the Δ*Aogcs1* and Δ*Aogts1* mutants were significantly reduced (*p* < 0.05) compared with the WT strain (Figure 5A). The RGI values for the WT (3.6%, 4.5%, and 19.8%), Δ*Aogcs1* (9.5%, 16.4%, and 44.9%), and Δ*Aogts1* (17.2%, 11.8%, and 27.8%) strains were calculated after incubation on TG plates at 34, 38, and 42 °C, respectively (Figure 5B).

The role of Arf-GAPs in cAMP–PKA signaling in *A. oligospora* was tested, and the Δ*Aogts1* mutant showed sensitivity to the PKA inhibitor, H-89, at a concentration of 10 µM (Appendix A). In addition, the transcriptional levels of the downstream genes of the cAMP/PKA, TOR, and MAPK pathways were significantly downregulated (*p* < 0.05) in the Δ*Aogcs1* and Δ*Aogts1* mutants on day 5 (Figure 5C).

### 3.7. AoGcs1 and AoGts1 Regulate Intracellular Lipid Storage and Endocytosis

To examine the roles of the *Aogcs1* and *Aogts1* genes in lipid utilization and endocytosis, we visualized LDs and endocytosis in fungal mycelia using BODIPY and an endocytosis tracer, the lipophilic styryl dye FM4-64, respectively. Compared with their distribution in the WT, LDs were remarkably reduced and unevenly distributed in the hyphae of the Δ*Aogcs1* mutant, whereas there was no difference in the distribution between the Δ*Aogts1* mutant and the WT strain. Moreover, it can be observed that the volumes of LDs in the two mutant strains were larger than the WT strain (Figure 6A). The FM4-64 dye was gradually internalized by the cells over time, and after 3 min of incubation, endosomes and vacuoles in the hyphae of the WT strain were almost completely stained. In contrast, only the plasma membrane near the hyphal wall and septa were stained by FM4-64 in the Δ*Aogcs1* and Δ*Aogts1* mutants after 3 min of incubation (Figure 6B). When the LDs in the WT and mutant cells were observed by TEM, we found that the LDs in the Δ*Aogcs1* and Δ*Aogts1* mutant strains were larger (Figure 6C). Statistical analysis showed that the diameter of LDs in Δ*Aogcs1* and Δ*Aogts1* mutant cells were approximately 2–3 times larger than the WT strain (Figure 6D).

### 3.8. Loss of Aogcs1 and Aogts1 Affects the ER Stress Response and Mitochondrial Activity

We examined the sensitivity of the Δ*Aogcs1* and Δ*Aogts1* mutant to DTT and tunicamycin, which induces ER-stress, and found that the Δ*Aogcs1* and Δ*Aogts1* mutants were hypersensitive to DTT (Figure 7A,B). In addition, mitochondria were unevenly distributed in the Δ*Aogcs1* and Δ*Aogts1* mutants and their morphology had changed (Figure 7C). Moreover, we determined mitochondrial activity using JC-1, a cationic dye that accumulates in energized mitochondria [56], and the mycelia of Δ*Aogcs1* and Δ*Aogts1* mutants displayed stronger green FI than the WT strain (Figure 7C). The ratio of the green/red FI ratio was significantly increased in the Δ*Aogcs1* and Δ*Aogts1* mutants when compared with the WT strain. The FI ratios of WT and mutant strains (i.e., Δ*Aogcs1* and Δ*Aogts1*) were 0.23, 0.34, and 0.45, respectively (Figure 7D).

### 3.9. Deletion of Aogcs1 and Aogts1 Results in ROS Accumulation and Enhanced Autophagy in Hyphae

The hyphal production of ROS was compared between WT and mutant strains. ROS were aggregated in certain regions of the mycelium in the WT strain, whereas ROS were scattered and dispersed in the mycelia of mutants (Figure 8A). The FI values for DHE in the WT and mutant strains (i.e., Δ*Aogcs1* and Δ*Aogts1*) were 44.6, 59.7, and 54.7, respectively (Figure 8B).

When MDC-stained hyphae of the WT and mutant strains were observed, the Δ*Aogcs1* and Δ*Aogts1* mutants produced more autophagosomes than the WT mutant strains (Figure 8A). The FI values for the WT, Δ*Aogcs1*, and Δ*Aogts1* strains were 40.4, 62.7, and 82.9, respectively (Figure 8C). Moreover, we observed more autophagosomes in the Δ*Aogcs1* and Δ*Aogts1* mutants than in the WT strain by TEM (Figure 8D). The transcriptional levels of five autophagy-related genes (i.e., *atg1*, *atg8*, *atg9*, *atg13*, and *atg17*) were upregulated in the mycelium of the Δ*Aogts1* mutant on day 7, whereas expression was upregulated in the Δ*Aogcs1* mutant on day 3 and not significantly different (*p* < 0.05) from the WT strain on day 7 (Figure 8E).

## 4. Discussion

Arf is a subfamily of small GTPase proteins that are ubiquitous and highly conserved in all eukaryotic cells, with more than one member present in every organism studied to date [57]. Arf-GAPs can switch from active Arf-GTP to inactive Arf-GDP by hydrolyzing GTP bound to Arf, and it plays a role in vegetative growth, sporulation, and the pathogenicity of fungi [18]. However, little is known about the functions of Arfs and Arf-GAPs in filamentous fungi. Here, two orthologous Arf-GAP proteins, AoGcs1 and AoGts1, were characterized in the NT fungus *A. oligospora*, and our results showed that AoGcs1 and AoGts1 are key regulators involved in diverse cellular functions in *A. oligospora*.

Previous studies have shown that Arf-GAPs are involved in the vegetative growth and development of hyphae [14,19,58]. In *C. albicans*, deletion of *age3* (homologous to *gcs1*) results in defective hyphal growth and loss of invasive hyphae [17]. Similarly, deletion of Moglo3 caused distinct growth defects on different media, suggesting that this protein was required for normal fungal growth of *M. oryzae* [18]. Recently, an orthologous Glo3 (AoGlo3) was identified for *A. oligospora*, and deletion of the *Aoglo3* gene resulted in growth defects and an increase in the hyphal septum. Furthermore, deletion of *gts1* in *A. gossypii* resulted in severe growth retardation as well as hyphae with spherical swelling and irregular branching [19]. In this study, we also found that Δ*Aogcs1* and Δ*Aogts1* mutants exhibited growth defects, and hyphae of the Δ*Aogts1* mutant became swollen compared to the WT; additionally, deletion of *Aogcs1* and *Aogts1* resulted in DNA damage and cell apoptosis. Thus, Arf-GAPs play a conserved role in hyphal growth and development in *A. oligospora* and other fungi.

The growth of the Δ*Aogcs1* and Δ*Aogts1* mutants was significantly inhibited by chemical stressors such as oxidants, cell wall-disturbing agents, and hypertonic reagents. Specifically, deletion of *Aogts1* caused severe damage to the cell wall of *A. oligospora*, which manifested as increased sensitivity to low concentrations of SDS (0.01%) and Congo red at any concentration, and was associated with a phenotype of altered hyphal and spore morphology. This result was further confirmed by increased expression of the genes involved in cell wall synthesis, especially *chs2*. Similarly, Δ*Moglo3* mutants have increased sensitivity to three cell wall inhibitors, namely, CFW, Congo red, and SDS [18], and the Δ*age3* mutant strain of *C. albicans* was susceptible to SDS [17]. In addition, Δ*Aogcs1* and Δ*Aogcs1* mutants were more sensitive to the oxidants menadione and H_2_O_2_, which may be related to the downregulation of some genes related to oxidative stress (*thr*, *glr*, and *thxR*) during the vegetative growth stage. Interestingly, different concentrations of the hypertonic reagents NaCl and sorbitol had different effects on the growth of the Δ*Aogcs1* and Δ*Aogts1* mutants, and the expression of osmotic stress-related genes was significantly upregulated, especially *msn2*. Moreover, deletion of *Aogcs1* and *Aogcs1* caused defective growth after heat shock. MSN2/4 are zinc finger protein transcription factors, which are involved in transcriptional induction via stress/cAMP-response elements, functioning in temperature, osmotic, oxidative, and hydrostatic pressure stress responses in *S. cerevisiae* [59,60]. In the entomopathogenic fungi *Beauveria bassiana* and *Metarhizium robertsii*, both Δ*Bbmsn2* and Δ*Mrmsn2* mutants lost 20–65% of their tolerance to hyperosmolarity, oxidation, carbendazim, cell wall perturbation, and high temperature at 34 °C during colony growth [61,62]. These results show that the orthologs of Gts1 and Gcs1 play a crucial role in the response of yeasts and filamentous fungi to environmental stress.

Asexual sporulation is crucial for the reproduction and survival of *A. oligospora* and other NT fungi [42]. In this study, the deletion of *Aogts1* caused a remarkable reduction in conidial yield, spore germination rate, and altered conidial morphology, whereas the Δ*Aogcs1* mutant showed a WT-like phenotype for conidiation. The marked reduction in conidia yield in the Δ*Aogts1* mutant was consistent with the altered expression of *Aogts1* during the sporulation stage, suggesting that *Aogts1* is critical for conidia production. In addition, the expression of several conidia-related genes, such as *abaA*, *fluG*, and *velB*, were significantly downregulated (*p* < 0.05) in the Δ*Aogts1* mutant during the sporulation stage. Previous studies have proved that *abaA*, *fluG*, and *velB* are important genes involved in conidiation of *Aspergillus nidulans* and other filamentous fungi [42,63,64]. Deletion of *velB* in *A. oligospora* abolished sporulation [42]. Similarly, conidia production was completely abolished in the *abaA* deletion mutant of *Fusarium graminearum* [65]. Additionally, both arf1 and arf2 are required for sporulation in *M. circinelloides* [66]. Moreover, similar results were reported in *M. oryzae*, where the Δ*Moglo3* mutant produced significantly fewer conidia compared to the WT strain [18]. These results suggest a conserved role for Arf and Arf-GAPs in regulating asexual development in *A. oligospora* and other filamentous fungi.

The NT fungi capture nematodes by producing traps and then decompose them for nutrients by secreting extracellular proteases [28]. Disruption of *Aogcs1* and *Aogts1* in *A. oligospora* resulted in a significant reduction in the number of traps followed by a significant decrease in nematode predation efficiency. The extracellular proteolytic activity of the Δ*Aogcs1* mutant was significantly increased (*p* < 0.05), but the proteolytic activity of the Δ*Aogts1* mutant was similar to WT, a result that was confirmed by significantly decreased (*p* < 0.05) transcription of the genes encoding serine proteases on day 5. These findings are consistent with results for *C. albicans* and *M. oryzae*. Knockout of *Caage3* results in the disruption of hyphal cell wall integrity and produces serious defects in invasive growth [17]. Likewise, the deletion of *glo3* in *M. oryzae* results in severe invasive growth defects and reduced virulence in the host [18]. These results suggest a conserved role for Arf-GAPs in infectious structure development, extracellular protease production, and fungal pathogenicity.

Arfs and Arf-GAPs are key components during vesicle trafficking and have been implicated in fungal growth and virulence [16,18,67]. Earlier studies found that the Arf-like protein yArl3 is required for protein transport from the ER to the Golgi or from the Golgi to the vacuole in *S. cerevisiae* [68]. Glo3 and Gcs1 mediate retrograde vesicular trafficking from the Golgi to the ER in *S. cerevisiae*, so mutant strains lacking both Glo3 and Gcs1 failed to proliferate, experienced a dramatic accumulation of ER, and were defective in protein trafficking between the ER and the Golgi [14]. The deletion of *gts1* in *A. gossypii* results in defective actin cytoskeleton organization and endocytosis [19]. In addition, Δ*Moglo3* mutants are defective in endocytosis, as well as in response of *M. oryzae* to ER stress and scavenging ROS [18]. Consistent with previous findings, Δ*Aogcs1* and Δ*Aogts1* mutants delayed uptake of the lipophilic dye FM4-64, exhibited increased sensitivity to ER stressors, and had significantly increased ROS levels in hyphae. These results suggest that Arf and Arf-GAPs play conserved functions in vesicle trafficking, endocytosis, ROS scavenging, and ER stress responses in fungi.

Previous studies have shown that Arf-GAP1 acts as a lipid-packaging sensor and helps anchor Arf-GAP1 to the highly curved membrane surface, allowing GTP hydrolysis on Arf1 [69]. In addition, Arf-GAPs are essential components of cellular signaling cascades that regulate the actin cytoskeletal network and gene expression changes, normal cell proliferation, survival through cell adhesion, and growth factor stimulation [70]. In this study, we found that the LDs in the mutants had increased in volume and were unevenly distributed in hyphal cells. Both Δ*Aogcs1* and Δ*Aogts1* mutants showed sensitivity to H-89, suggesting that the activities of PKA signaling pathways were reduced or lost in the mutant strains. The transcriptional downregulation of the genes related to the PKA and TOR signaling pathway further verified the result that Arf-GAPs are involved in intracellular signal transduction. In addition, cAMP/PKA signaling has been proved to be involved in conidiation, secondary metabolism, and pathogenicity in diverse fungi [71]. Taken together, these results indicate that Arf-GAPs are important multifunctional regulators involved in membrane trafficking, coordination of lipid homeostasis, and intracellular signaling in *A. oligospora* and other fungi.

Arf regulates various membrane functions, including ER stress and the ER-mitochondrial encounter structure [11,72]. Mitochondria and ER are highly interacting organelles that contribute to hyphal growth, mitochondrial morphology, and virulence in the pathogenic mold *Aspergillus fumigatus* [73]. ATP production, signaling, apoptosis, and autophagy are all regulated by multifunctional regulators in the mitochondria [74]. In *S. cerevisiae* overexpressing Gcs1, the mitochondria were morphologically abnormal and functionally impaired, with unbranched tubules and large spherical structures; therefore, it is thought that Gcs1 may participate in the maintenance of mitochondrial morphology by organizing the actin cytoskeleton [75]. Our previous study demonstrated that deletion of *glo3* enhanced autophagy in *A. oligospora* [51]. Similarly, our present study found that knockout of *Aogts1* and *Aogcs1*, apart from affecting the ER stress response, DNA damage, and apoptosis, also significantly reduced MMP, which indicates impaired mitochondrial activity. In addition, the enhanced MDC fluorescence intensity, a significantly reduced green-to-red fluorescence ratio after JC-1 staining, upregulated autophagy-related gene expression, and increased autophagosomes observed by TEM in the mutants further indicate that Arf-GAPs play an essential role in the regulation of fungal autophagy and mitochondrial activity.

In this study, we identified two orthologous Arf-GAPs in *A. oligospora*. Our findings suggest that AoGcs1 and AoGts1 play multifunctional roles in the growth, development, and pathogenicity of this fungus. Knockout of *Aogcs1* and *Aogts1* in *A. oligospora* affected intracellular signaling associated with cAMP/PKA signaling, which regulates downstream transcription factors (such as *Msn2*), resulting in increased sensitivity to intracellular and extracellular stressors. The deletion of *Aogcs1* and *Aogts1* also altered the morphology and function of the nucleus and inhibited intracellular vesicle trafficking. Abnormal nuclear function causes an ER stress response, and defective ER function impairs the exchange and renewal of the membrane system between ER–Golgi and ER–mitochondria, resulting in a decrease in MMP and an increase in intracellular ROS and enhanced autophagy. In addition, the remarkable accumulation of LDs is indicative of Golgi deficiency, and the abnormally increased ROS will aggravate DNA damage. This combination results in slow growth of the fungus, increased sensitivity to stressors, and decreased sporulation and pathogenicity (Figure 9).

## 5. Conclusions

Our work showed that AoGcs1 and AoGts1 play a similar role in the growth, development, and pathogenicity of *A. oligospora*. These Arf-GAP proteins are involved in the regulation of DNA damage, endocytosis, ROS degradation, ER stress response, lipid storage, mitochondrial activity, and autophagy, and their deletion impairs mycelial growth, conidiation, stress responses, and nematode predation efficiency. Our results have expanded the understanding of the biological functions of Arf-GAP proteins in *A. oligospora* and provide novel insights into the role of Arf-GAPs in pathogenic fungal growth and development.

## Figures and Tables

**Figure 1 jof-08-00463-f001:**
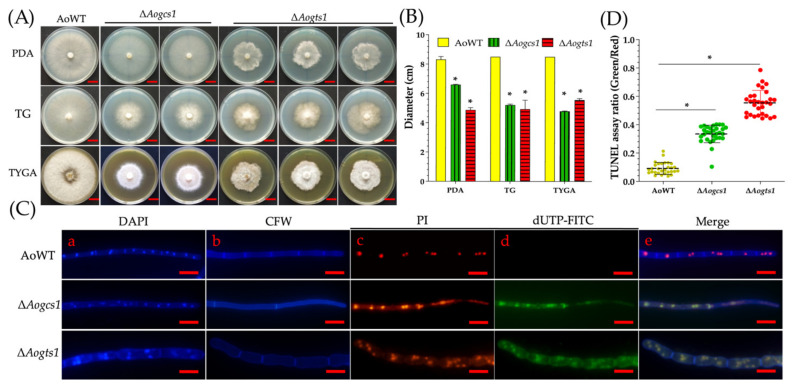
Comparison of colony morphology, mycelial growth, DNA damage, and cell apoptosis in the wild-type (WT) and mutant strains (i.e., Δ*Aogcs1* and Δ*Aogts1*) of *A. oligospora*. (**A**) Colonies from the WT and mutant strains were cultured on PDA, TG, and TYGA plates for 7 days at 28 °C. Bar = 2 cm. (**B**) Colony diameters of the WT and mutant strains cultured on different media for 7 days. (**C**) Comparison of nuclei, cell morphology, and DNA damage. (**a**,**b**) Mycelia were stained with 4′,6-diamidino-2-phenylindole (DAPI, (**a**)) and calcofluor white (CFW, (**b**)); (**c**) Nuclei of the hyphae stained with propidium iodide (PI); (**d**) Free DNA of the hyphae re-stained with FITC-dUTP. (**e**) Merged images from panels (**c**–**e**). Bar = 10 μm. (**D**) Analysis of DNA fragmentation and cell apoptosis in hyphal cells. The ratio of green to red fluorescence intensity was determined for at least 30 fields observed under a microscope. Asterisk: significant difference between mutant and WT (Tukey’s HSD, *p* < 0.05).

**Figure 2 jof-08-00463-f002:**
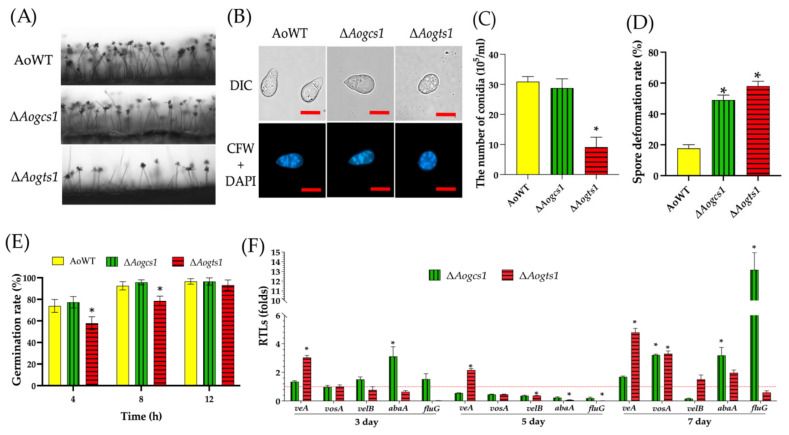
Comparison of conidiation, conidial germination, conidial morphology, and relative transcriptional levels (RTLs) of sporulation-related genes between WT and mutant strains of *A. oligospora*. (**A**) Sporulation in the WT and mutant strains; (**B**) conidia of WT and mutant strains were stained with CFW and DAPI (bar = 10 μm); (**C**) comparison of conidial yields between the WT and mutant strains; (**D**) spore deformation rate of WT and mutant strains; (**E**) conidial germination rates in WT and mutant strains; (**F**) comparison of RTLs of sporulation-related genes in the mutant strain and WT strain at different time points. The red line indicates the standard (which has an RTL of 1) for statistical analysis of the RTL of each gene in a deletion mutant compared to that in the WT strain. The asterisk indicates a significant difference between the mutants and the WT strain (Tukey’s HSD, *p* < 0.05).

**Figure 3 jof-08-00463-f003:**
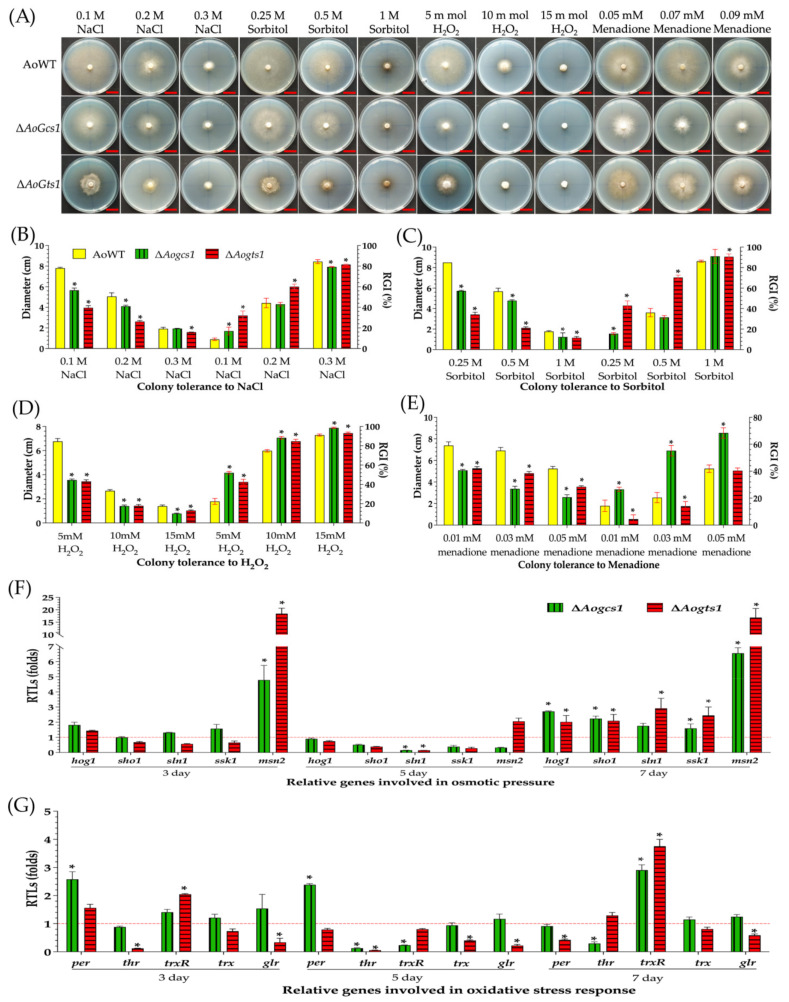
Comparison of the stress tolerance ability of the WT and mutant strains to oxidative stress and osmotic pressure. (**A**) Colony morphologies of the WT and mutant strains under oxidative stress and osmotic pressure. Bar = 2 cm. (**B**–**E**) Colony diameter and relative growth inhibition (RGI) values for strains under oxidative and osmotic pressure. (**F**) Relative transcription levels (RTLs) of genes involved in the osmotic stress response in the mutants when compared with the WT strain at different time points. (**G**) RTLs of genes involved in the oxidative stress response in the mutants when compared with the WT strain at different time points. The red line (**F**,**G**) indicates the standard (which has an RTL of 1) for statistical analysis of the RTL of each gene in a deletion mutant compared to that in the WT strain. The asterisk indicates a significant difference between the mutants and the WT strain (Tukey’s HSD, *p* < 0.05).

**Figure 4 jof-08-00463-f004:**
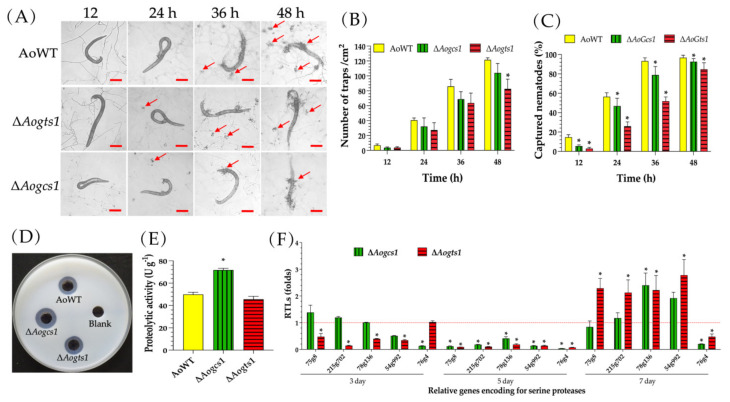
Comparison of trap formation, nematocidal activity, and extracellular proteolytic activity in the WT and mutant strains. (**A**) Trap formation in the WT and mutant strains induced by nematodes at different time points. The red arrows point towards the traps produced by the WT strain and mutants. Bar = 100 μm. (**B**) The number of traps produced by the WT and mutant strains. (**C**) The percentage of captured nematodes at different time points. (**D**) Comparison of the extracellular proteolytic activities on casein plates. (**E**) Total extracellular proteolytic activity of the WT and mutant strains exhibited in 7-day old PD broth. (**F**) Relative transcription levels (RTLs) of the genes encoding serine proteases in the mutants when compared with the WT strain at different time points. The red line indicates the standard (which has an RTL of 1) for statistical analysis of the RTL of each gene in a deletion mutant compared to that in the WT strain. The asterisk indicates a significant difference between the mutants and the WT strain (Tukey’s HSD, *p* < 0.05).

**Figure 5 jof-08-00463-f005:**
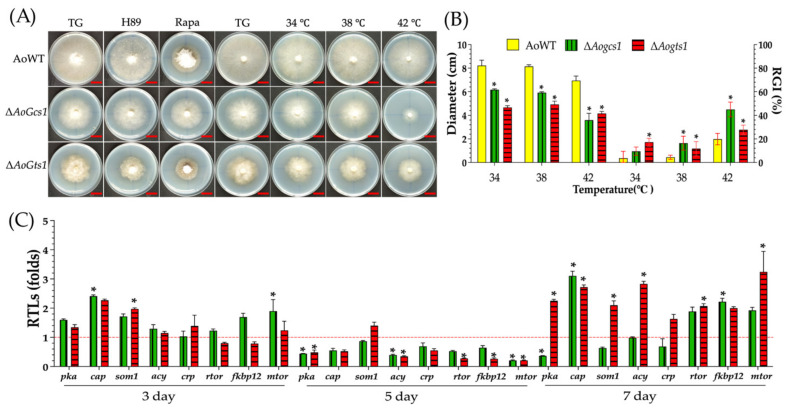
Comparison of rapamycin, H89, and heat shock sensitivity in the WT and mutant strains. (**A**) Colony morphologies of the WT and mutant strains cultured in the presence of 10 ng/mL rapamycin or 10 µM H-89. Bar = 2 cm. (**B**) Colony diameters and the relative growth inhibition (RGI) values for the WT and mutant strains cultured at a temperature of 34–42 °C. (**C**) Relative transcription levels (RTLs) of the cAMP/PKA, TOR, and MAPK signaling pathway-related genes in the mutants when compared with the WT strain at different time points. The red line indicates the standard (which has an RTL of 1) for statistical analysis of the RTL of each gene in a deletion mutant compared to that in the WT strain. The asterisk indicates a significant difference between the mutants and the WT strain (Tukey’s HSD, *p* < 0.05).

**Figure 6 jof-08-00463-f006:**
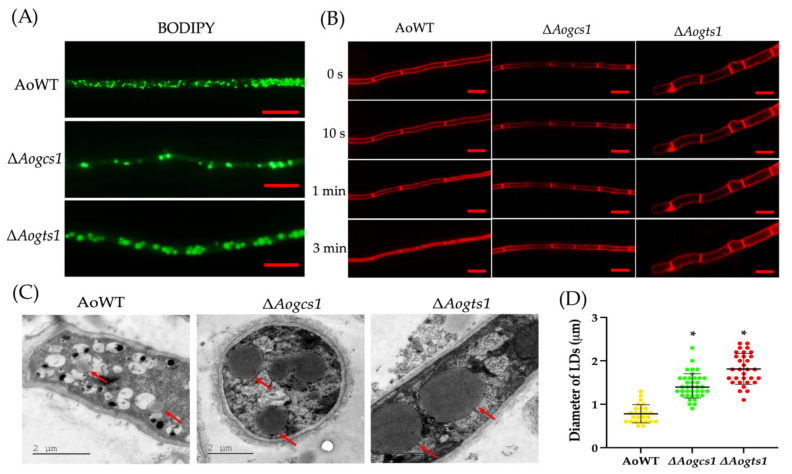
Effects of AoGcs1 and AoGts1 on lipid storage and endocytosis in mycelia. (**A**) Mycelia of the WT and mutant strains were stained with the fluorescent dye BODIPY. Bar: 10 = μm. (**B**) Endocytosis in the WT and mutant strains at different times. Bar: 10 = μm. (**C**) Lipid droplets (LDs) were observed in the WT and mutant strains by transmission electron microscope. The arrows show the LDs. (**D**) Comparison of the diameters of the LDs in more than 30 cells in electron micrographs of mutant strains and the WT strain. The asterisk indicates a significant difference between the mutants and the WT strain (Tukey’s HSD, *p* < 0.05).

**Figure 7 jof-08-00463-f007:**
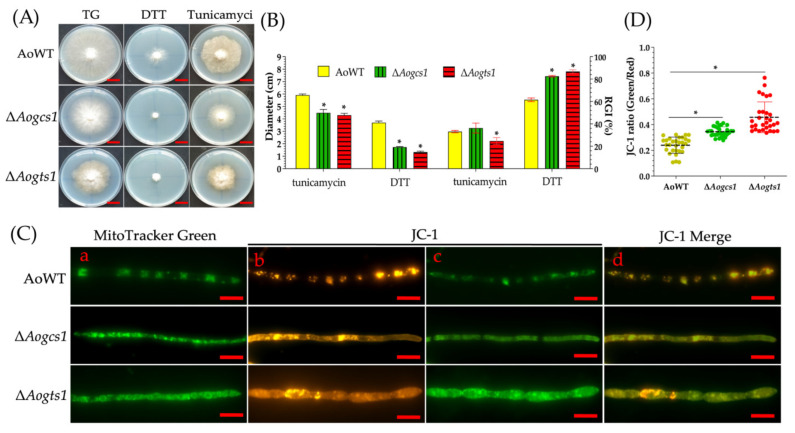
Influence of AoGcs1 and AoGts1 on the endoplasmic reticulum stress response and on mitochondrial morphology and activity. (**A**) Colony morphologies of WT and mutant strains cultured in the presence of DTT and tunicamycin at a concentration of 10 mM. Bar = 2 cm. (**B**) Colony diameters and the relative growth inhibition (RGI) values of the WT and mutant strains cultured with 10 mM DTT and 10 mM tunicamycin. (**C**) Comparison of mycelial mitochondrial morphology and mitochondrial membrane potential (MMP) of WT and mutant strains: (**a**) mycelia were stained with MitoTracker Green; (**b**–**d**) MMP was analyzed with tetraethylbenzimidazolyl-carbocyanine iodide (JC-1) staining. Bar = 10 μm. (**D**) MMP level analysis in mycelia. The ratio of green to red fluorescence intensity was obtained for at least 30 fields viewed under a microscope. The asterisk indicates a significant difference between the mutants and the WT strain (Tukey’s HSD, *p* < 0.05).

**Figure 8 jof-08-00463-f008:**
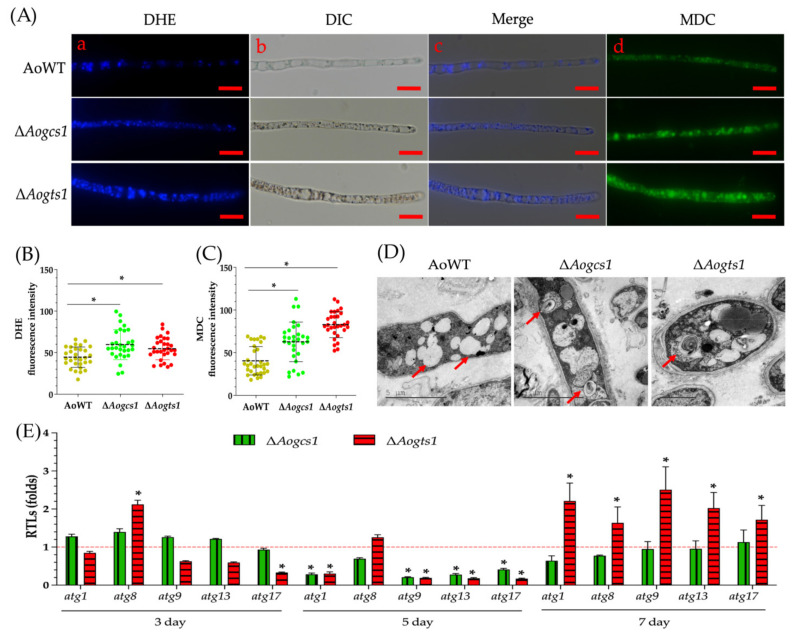
Effect of AoGcs1 and AoGts1 on ROS levels and autophagy in mycelia. (**A**) Mycelia of the WT and mutants were stained with dihydroethidium (DHE; (**a**–**c**)) and monodansylcadaverine (MDC; (**d**)). Bar = 10 μm. (**B**) The ROS levels of the WT and mutant strains were analyzed by estimating the DHE fluorescence intensity values for at least 30 fields. (**C**) Autophagic activity was analyzed by estimating the MDC fluorescence intensity values for at least 30 fields viewed under a microscope. (**D**) The autophagosomes observed by transmission electron microscope (TEM). Red arrows indicate autophagosomes. (**E**) Relative transcription levels (RTLs) of autophagy-related genes were determined at different time points. The red line indicates the standard (which has an RTL of 1) for statistical analysis of the RTL of each gene in a deletion mutant compared to that in the WT strain. The asterisk indicates a significant difference between the mutants and the WT strain (Tukey’s HSD, *p* < 0.05).

**Figure 9 jof-08-00463-f009:**
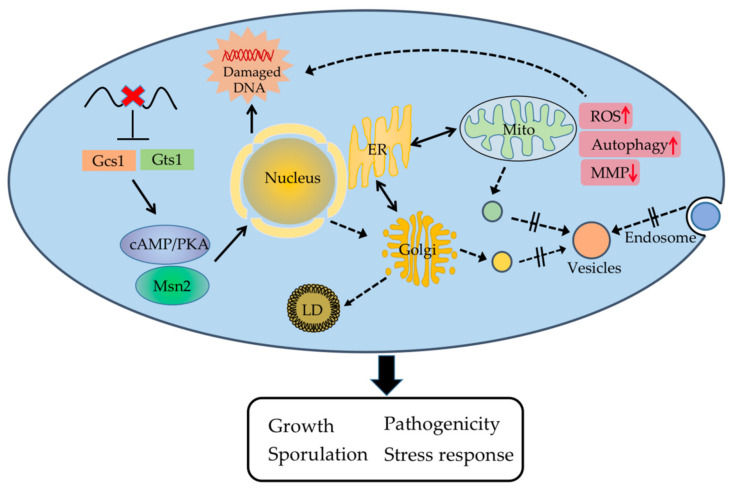
Schematic diagram of the regulation of AoGcs1 and AoGts1 in *A*. *oligospora*. In *A. oligospora*, AoGcs1 and AoGts1 are involved in intracellular signaling related to cAMP/PKA, which further regulates related transcription factors and plays a key role in vesicle trafficking, nuclear morphology, apoptosis, ER stress, mitochondrial activity, ROS scavenging, autophagy, and lipid droplet storage, ultimately regulating fungal phenotypic traits. LD, lipid droplet; Mito, mitochondria; ER, endoplasmic reticulum; MMP, mitochondrial membrane potential.

## Data Availability

Not applicable.

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
