# Peer review of "The Arf-GAP Proteins AoGcs1 and AoGts1 Regulate Mycelial Development, Endocytosis, and Pathogenicity in Arthrobotrys oligospora"

_jof, 2022, doi:10.3390/jof8050463_

Round 1

Reviewer 1 Report

The authors have made great efforts to characterize the Arf-GAP proteins AoGcs1 and AoGts1 in A. oligospora by deletion of Aogcs1 and Aogts1 genes. The attained results elucidated the role of Arf-GAPs in mycelial development, endocytosis, and pathogenicity of the nematophagous fungi. The results are of interest to readers and micrologists. Generally, the manuscript has been well documented and organized. However, some weak points have been found and should be revised and edited to improve the quality of the paper. You can find some useful comments and suggestions (see in detail in pdf file).

The objectives of this study should be specifically narrated (line 93-94)

The part of Fungal strains and culture conditions, more information must be provided (line 101-106)

The numbering of subsection must be checked and changed in order, they were duplicated

Line 170-171, or heat shock, why the 28,34,38 and 42oC were selected? the grown for 28oC for 7 days ?. This method was developed by you or other authors?

All Fig should be rechecked and add the scale bar (see in pdf file).

Conclusion part must be reworded and should narrate the most significant findings in this study

Check the duplicated sentences in whole text, and grammars, typing errors etc. All names of genes and scientific name of fungi must be italic (see in pdf file).

One of the weakest points of the paper is too high similarity rate (over 52% in all). Critically reduce as much as possible (25-30%). 

Author Response

Response: Thank you for your valuable comments. We are very appreciative of your great help to improve our manuscript. We have carefully considered and revised the manuscript according to your comments.

The objectives of this study should be specifically narrated (line 93-94)

Response: The objectives of this study have been revised according to your suggestion.

The part of Fungal strains and culture conditions, more information must be provided (line 101-106)

Response: Thank you. We have revised this paragraph according to your suggestion.

The numbering of subsection must be checked and changed in order, they were duplicated.

Response: Thank you. We have checked and revised the numbering of subsection.

Line 170-171, or heat shock, why the 28, 34, 38 and 42oC were selected? the grown for 28oC for 7 days ?. This method was developed by you or other authors?

Response: This method was developed by our previous study, we have added related reference. 

All Fig should be rechecked and add the scale bar (see in pdf file).

Response: We have added the scale bar according to your suggestion.

Conclusion part must be reworded and should narrate the most significant findings in this study

Response: Thank you for your suggestion. We have revised the Conclusion part.

Check the duplicated sentences in whole text, and grammars, typing errors etc. All names of genes and scientific name of fungi must be italic (see in pdf file).

Response: Thank you for your suggestion. We have carefully revised our manuscript according to your comments (in pdf file).

One of the weakest points of the paper is too high similarity rate (over 52% in all). Critically reduce as much as possible (25-30%).

Response: Thank you for your suggestion. We have revised the manuscript to reduce the similarity rate.

Reviewer 2 Report

Special comments

  • Specific comments have been made in the text

Author Response

It is better to put the name of the authors and put the website in the references (i.e Gasteiger et al., 2005; Bjellqvist et al., 1993; 1994).

Response: We have added the related reference.

idem

Response: We have added the related reference.

Reviewer 3 Report

The manuscript (The Arf-GAP Proteins AoGcs1 and AoGts1 Regulate Mycelial Development, Endocytosis, and Pathogenicity in Arthrobotrys oligospora) is interesting but substantial revision is necessary before publication.

Please re-write the aim of study at the end of the introduction concentrating on the objectives of the current research.

Some methods should be explained in details such as Quantitative Real-Time PCR (RT-PCR).

Please highlight your results in the conclusion section.

Several errors were found and the manuscript should be carefully revised for English and grammatical errors.

L50 Drive should be drives

L55 Please change to of proteins

L57 Please change to GTP hydrolysis

L87 Change the sentence to (Since the genome of A. oligospora was sequenced in 2011, studies have focused on regulating mycelia growth, development, and pathogenicity of A. oligospora).

L104 For screening of should be to screen.

L105 construct should be constructed.

L226 Please change Janpa to Japan.

L292 Please change to (However, no difference was observed …….).

L333 Delete all.

L428 Replace were with was.

L449 Were should be was.

L453 Change to (The green/red FI ratio …..).

L509 It should be the hyphal septum.

L517 I think you mean cell wall perturbation or perturbance, correct please.

L521 Correct to the significance.

L549 Correct to the condition.

L604 Correct were to was.

L611 Correct indicating to indicates.

L617 Should be the growth.

Author Response

Response: Thank you for your positive comments. We are very appreciative of your great help to improve our manuscript. We have carefully considered and revised the manuscript according to your comments.

Please re-write the aim of study at the end of the introduction concentrating on the objectives of the current research.

Response: Done. We have rewritten the aim of study at the end of the introduction.

Some methods should be explained in details such as Quantitative Real-Time PCR (RT-PCR).

Response: Done. We have explained in details the method of Quantitative Real-Time PCR (RT-PCR).

Please highlight your results in the conclusion section.

Response: Done. We have highlighted your results in the conclusion section.

Several errors were found and the manuscript should be carefully revised for English and grammatical errors.

L50 Drive should be drives

Response: Done.

L55 Please change to of proteins

Response: Done.

L57 Please change to GTP hydrolysis

Response: Done.

L87 Change the sentence to (Since the genome of A. oligospora was sequenced in 2011, studies have focused on regulating mycelia growth, development, and pathogenicity of A. oligospora).

Response: Done.

L104 For screening of should be to screen.

Response: Done.

L105 construct should be constructed.

Response: Done.

L226 Please change Janpa to Japan.

Response: Done.

L292 Please change to (However, no difference was observed …….).

Response: Deletion of Aogts1 resulted in a remarkable reduction in conidiophores, so we keep the origin sentence “However, few conidiophores were observed…….”.

L333 Delete all.

Response: Done.

L428 Replace were with was.

Response: Done.

L449 Were should be was.

Response: Done.

L453 Change to (The green/red FI ratio …..).

Response: Done.

L509 It should be the hyphal septum.

Response: Done.

L517 I think you mean cell wall perturbation or perturbance, correct please.

Response: Done.

L521 Correct to the significance.

Response: Done.

L549 Correct to the condition.

Response: It is conidiation, not condition.

L604 Correct were to was.

Response: Done.

L611 Correct indicating to indicates.

Response: Done.

L617 Should be the growth.

Response: Done.

Reviewer 4 Report

The Manuscript entitled (The Arf-GAP Proteins AoGcs1 and AoGts1 Regulate Mycelial Development, Endocytosis, and Pathogenicity in Arthrobotrys oligospora) provide novel insights into the role of Arf-GAPs in pathogenic fungal growth and development. The manuscript has many data and good results those are introduced very well in all sections. Here some comments for the authors that are considered as minor revision

  • Lines 94 to 98. This paragraph is results. Thus, please write here clear and in details the objectives of the paper
  • Line 177: Approximately OR exact ? How if Approximately to quantify the number of traps and captured nematodes
  • Line 254: (compared to zero time) instead of (compared to 0 h)
  • In results, Please write the values of (p < 0.05) in exact value as, for example (p = 0.03). Also. Add F values and df for all results.

Author Response

Response: Thank you for your valuable comments. We have carefully considered and revised the manuscript according to your comments.

Lines 94 to 98. This paragraph is results. Thus, please write here clear and in details the objectives of the paper

Response: Thank you for your suggestion. We have revised these sentences.

Line 177: Approximately OR exact ? How if Approximately to quantify the number of traps and captured nematodes

Response: we have revised this sentence.

Line 254: (compared to zero time) instead of (compared to 0 h)

Response: we have revised this sentence.

In results, Please write the values of (p < 0.05) in exact value as, for example (p = 0.03). Also. Add F values and df for all results.

Response: Thank you for your suggestion. We used the p < 0.05 as a threshold value for considering a significant difference, because there are multiple genes or mutants were analyzed, and their p values are varied in different genes. So we keep the original description.

Round 2

Reviewer 1 Report

The revised paper has been significantly made following the comments and suggestions of the reviewers. However, some minor revisions still need as below:

+ In section 2.1. Fungal strains and culture conditions, where did the authors collect the ATCC24927 and derived mutants, and what kinds of the derived mutants?

+ Conclusion should be further improved,  one or two sentences which are the most significant results of this study should be narrated.

Author Response

+ In section 2.1. Fungal strains and culture conditions, where did the authors collect the ATCC24927 and derived mutants, and what kinds of the derived mutants?

Response: We have added the related information according to your suggestion.

+ Conclusion should be further improved, one or two sentences which are the most significant results of this study should be narrated.

Response: Thank you. We have revised the Conclusion according to your suggestion.

Reviewer 3 Report

Please consider improving the following points:

L95 Objectives: It is not acceptable to write any results in this part. I request you just explain your study's aim in specified points.

L254 Again please explain the details of RT-PCR. You did not add any details to the previous version.

L738 The conclusion should include a summary of your study results. More statements are needed to show the quality of your work.

Author Response

Please consider improving the following points:

L95 Objectives: It is not acceptable to write any results in this part. I request you just explain your study's aim in specified points.

Response: Thank you. We have revised the related sentence. 

L254 Again please explain the details of RT-PCR. You did not add any details to the previous version.

Response: We have explained the details of RT-PCR in the revised manuscript.

L738 The conclusion should include a summary of your study results. More statements are needed to show the quality of your work.

Response: Thank you. We have revised the Conclusion according to your suggestion.